# AvE: Assistance via Empowerment

**Yuqing Du**
UC Berkeley
yuqing_du@berkeley.edu

**Stas Tiomkin**
UC Berkeley
stas@berkeley.edu

**Emre Kıcıman**
Microsoft Research
emrek@microsoft.com

**Daniel Polani**
University of Hertfordshire
d.polani@herts.ac.uk

**Pieter Abbeel**
UC Berkeley
pabbeel@berkeley.edu

**Anca Dragan**
UC Berkeley
anca@berkeley.edu

## Abstract

One difficulty in using artificial agents for human-assistive applications lies in the challenge of accurately assisting with a person's goal(s). Existing methods tend to rely on inferring the human's goal, which is challenging when there are many potential goals or when the set of candidate goals is difficult to identify. We propose a new paradigm for assistance by instead increasing the *human's ability to control* their environment, and formalize this approach by augmenting reinforcement learning with *human empowerment*. This task-agnostic objective preserves the person's autonomy and ability to achieve any eventual state. We test our approach against assistance based on goal inference, highlighting scenarios where our method overcomes failure modes stemming from goal ambiguity or misspecification. As existing methods for estimating empowerment in continuous domains are computationally hard, precluding its use in real time learned assistance, we also propose an efficient empowerment-inspired proxy metric. Using this, we are able to successfully demonstrate our method in a shared autonomy user study for a challenging simulated teleoperation task with human-in-the-loop training.

## 1   Introduction

We aim to enable artificial agents, whether physical or virtual, to assist humans in a broad array of tasks. However, training an agent to provide assistance is challenging when the human's goal is unknown because that makes it unclear what the agent should do. Assistance games [18] formally capture this as the problem of working together with a human to maximize a common reward function whose parameters are only known to the human and not to the agent. Naturally, approaches to assistance in both shared workspace [32, 14, 13, 33, 31, 25] and shared autonomy [21, 20, 11, 15, 32] settings have focused on inferring the human's goal (or, more broadly, the hidden reward parameters) from their ongoing actions, building on tools from Bayesian inference [5] and Inverse Reinforcement Learning [30, 37, 2, 4, 45, 19, 34, 12, 26]. However, goal inference can fail when the human model is misspecified, e.g. because people are not acting noisy-rationally [27, 36], or because the set of candidate goals the agent is considering is incorrect [7]. In such cases, the agent can infer an incorrect goal, causing its assistance (along with the human's success) to suffer, as it does in Figure 1.

Even in scenarios where the agent correctly infers the human's goal, we encounter further questions about the nature of collaborations between humans and assistive agents: what roles do each of them play in achieving the shared goal? One can imagine a scenario where a human is attempting to traverse down a hallway blocked by heavy objects. Here, there are a range of assistive behaviours: a robot could move the objects and create a path so the human is still the main actor, or a robot could physically carry the person down the hallway, making the human passive. Depending on the context, either solution may be more or less appropriate. Specifically, the boundary between *assisting* humans

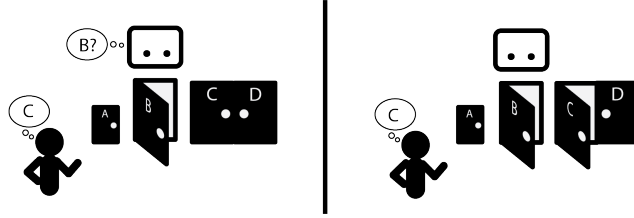

Figure 1: Toy scenario where a robot operates multiple doors. On the left, the robot attempts to infer the human's intended goal, $C$, but mistakenly infers $B$. On the right, the robot assesses that the human's empowerment would increase with doors $B$ and $C$ open. Naively opening all doors will not increase empowerment as $A$ is too small for the person and $D$ leads to the same location as $C$.

in tackling challenging tasks and *solving* these tasks in the place of humans is not clearly defined. As AI improves at 'human' jobs, it is crucial to consider how the technology can complement and amplify human abilities, rather than replace them [43, 42].

Our key insight is that agents can assist humans without inferring their goals or limiting their autonomy by instead increasing the human's *controllability* of their environment – in other words, their *ability to affect the environment through actions*. We capture this via *empowerment*, an information-theoretic quantity that is a measure of the controllability of a state through calculating the logarithm of the number of possible distinguishable future states that are reachable from the initial state [39]. In our method, Assistance via Empowerment (AvE), we formalize the learning of assistive agents as an augmentation of reinforcement learning with a measure of human empowerment. The intuition behind our method is that by prioritizing agent actions that increase the human's empowerment, we are enabling the human to more easily reach whichever goal they want. Thus, we are assisting the human without information about their goal – the agent does not carry the human to the goal, but instead clears a path so they can get there on their own. Without any information or prior assumptions about the human's goals or intentions, our agents can still learn to assist humans.

We test our insight across different environments by investigating whether having the agent's behaviour take into account human empowerment during learning will lead to agents that are able to assist humans in reaching their goal, despite having no information about what the goal truly is. Our proposed method is the first one, to our knowledge, that successfully uses the concept behind empowerment with real human users in a human-agent assistance scenario. Our experiments suggest that while goal inference is preferable when the goal set is correctly specified and small, empowerment can significantly increase the human's success rate when the goal set is large or misspecified. This does come at some penalty in terms of how quickly the human reaches their goal when successful, pointing to an interesting future work direction in hybrid methods that try to get the best of both worlds. As existing methods for computing empowerment are computationally intensive, we also propose an efficient empowerment-inspired proxy metric that avoids the challenges of computing empowerment while preserving the intuition behind its usefulness in assistance. We demonstrate the success of this algorithm and our method in a user study on shared autonomy for controlling a simulated dynamical system. We find that the strategy of stabilizing the system naturally emerges out of our method, which in turn leads to higher user success rate. While our method cannot outperform an oracle method that has knowledge of the human's goal, we find that increasing human empowerment provides a novel step towards generalized assistance, including in situations where the human's goals cannot be easily inferred.

**Main contributions:**

- We formalize learning for human-agent assistance via the empowerment method.

- We directly compare our method against a goal inference approach and confirm where our method is able to overcome potential pitfalls of inaccurate goal inference in assistance.

- We propose a computationally efficient proxy for empowerment in continuous domains, enabling human-in-the-loop experiments of learned assistance in a challenging simulated teleoperation task.

## 2 Empowerment Preliminary

**Background.** To estimate the effectiveness of actions on the environment, [23] proposed computing the *channel capacity* between an action sequence, $\vec{A}_T \doteq (A_1 A_2, \ldots, A_T)$, and the final states, $S_T$, where the channel is represented by the environment. Formally, empowerment of a state $s$ is:

$$\mathcal{E}(s) = \underset{p(\vec{A}_T|s)}{\text{maximum }} \mathcal{I}[\vec{A}_T; S_T \mid s], \tag{1}$$

where $\mathcal{I}[\vec{A}_T; S_T \mid s] \doteq \mathcal{H}(S_T \mid s) - \mathcal{H}(S_T \mid \vec{A}_T, s)$ is the mutual information between $\vec{A}_T$ and the $S_T$, $\mathcal{H}(\cdot)$ is entropy, and $T$ is the time horizon, which is a hyperparameter. The *probing probability distribution* $p(\vec{A}_T|s)$ is only used for the computation of the empowerment/channel capacity $\mathcal{E}(s)$, and never generates the behavior directly. Note that the actions are applied in an open-loop fashion, without feedback.

In the context of learning, empowerment as an information-theoretic quantity has mainly been seen as a method for producing intrinsic motivation [39, 29, 24, 10]. [17] showed that a composition of human empowerment, agent empowerment, and agent-to-human empowerment are useful for games where a main player is supported by an artificial agent. An alternative view of this compositional approach was proposed in [40], where the agent-to-human empowerment was replaced by human-to-agent empowerment. The latter is a human-centric approach, where only the humans' actions affect the agents' states in a way which is beneficial for the human. These approaches have conceptually discussed the applicability of empowerment to human-agent collaborative settings and building on this, our paper concretely proposes that assistance in shared workspaces and shared autonomy tasks can be cast as an empowerment problem, and evaluates that idea with real users.

Given the challenge of computing empowerment, some existing estimation methods are:

**Tabular case approximations.** In tabular cases with a given channel probability, $p(S_T \mid \vec{A}_T, s)$, the problem in (1) is solved by the Blahut-Arimoto algorithm [6]. This method does not scale to: a) high dimensional state and/or action space, and b) long time horizons. An approximation for a) and b) can be done by Monte Carlo simulation [22], however, the computational complexity precludes user studies of empowerment-based methods for assistance in an arbitrary state/action space.

**Variational approximations.** Previous work has proposed a method for using variational approximation to provide a lower bound on empowerment [29]. This was extended later to closed-loop empowerment [16]. Both of these methods estimate empowerment in a model-free setting. Recent work has also proposed estimating empowerment by the latent space water-filling algorithm [44], or by applying bi-orthogonal decomposition [41], which assumes known system dynamics. However, these estimation methods are computationally hard, which precludes their use in reinforcement learning, especially, when a system involves learning with humans, as in our work.

## 3 Assistance via Empowerment

### 3.1 Problem Setting

We formulate the human-assistance problem as a reinforcement learning problem, where we model assistance as a Markov Decision Process (MDP) defined by the tuple $(\mathcal{S}, \mathcal{A}_a, \mathcal{P}, R, \gamma)$. $\mathcal{S}$ consists of the human and agent states, $(S_h, S_a)$, $\mathcal{A}_a$ is the set of agent actions, $\mathcal{P}$ is an unknown transition function, $R$ is a reward function, and $\gamma$ is a discount factor. We assume the human has an internal policy $\pi_h$ to try to achieve a goal $g^* \in S_h$ that is unknown to the agent. Note that the human may not behave completely rationally and $\pi_h$ may not enable them to achieve $g^*$, requiring assistance. As the agent does not know the human policy, we capture state changes from human actions in the environment transition function $(s_h^{t+1}, s_a^{t+1}) = \mathcal{P}((s_h^t, s_a^t), a_a)$.

Our method uses reinforcement learning to learn an assistive policy $\pi_a$ that maximizes the expected sum of discounted rewards $\mathbb{E}[\sum_t \gamma^t R(s_a^t, s_h^t)]$. For assistance, we propose the augmented reward function

$$R(s_a, s_h) = R_{original}(s_a, s_h) + c_{emp} \cdot \mathcal{E}_{human}(s_h)$$

where $R_{original}$ captures any general parts of the reward that are non-specific to assisting with the human's goal, such as a robot's need to avoid collisions, and where $\mathcal{E}_{human}(s_h)$ is the agent's

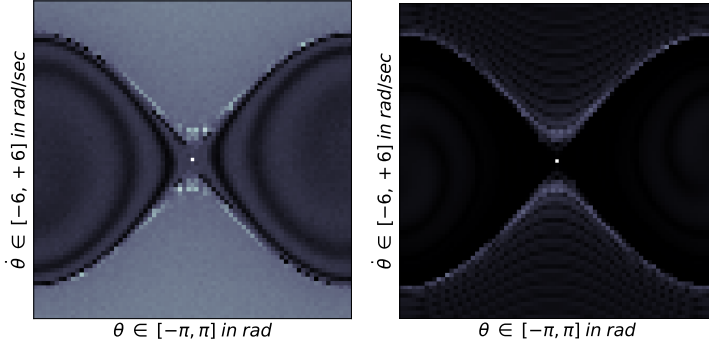

Figure 2: Comparison between our proxy for empowerment (left) and the known landscape (right) of the non-linear pendulum over a long time horizon. The proxy landscape captures the essential properties of the empowerment landscape: maximum at the upright position and comparatively low values for states with energy below the separatrix energy.

estimation of the human's empowerment in their current state, $s_h$. In simple tabular environments we can directly compute empowerment, however, this does not extend to more realistic assistance scenarios. Thus for our user study, we propose an empowerment-inspired proxy as detailed in Section 3.2. We access the human's action and state space either through observation in the shared workspace case or directly in shared autonomy. The coefficient $c_{emp} \geq 0$ is included to balance the weighting between the original reward function and the empowerment term.

## 3.2   Approach for Human-in-the-Loop Training

As our work is intended for real-time human assistance applications, we strongly prioritize computational efficiency over the numerical accuracy of the empowerment estimation in our user study. To compute true empowerment, one considers potential forward-simulations of agent actions of duration $T$ starting in $s_{init}$ and their resulting effect. However, since existing methods for approximating empowerment in continuous domains from empirical data are computationally quite hard and do not scale well, here we instead draw on the intuition of empowerment to propose a proxy metric using a measure of diversity of final states as a surrogate for the channel capacity.

Namely, we use a fast and simple analogy of the sparse sampling approximation from [38] for the empowerment method. In that work, the number of discrete states visited by the forward-simulations was counted. A large number of different final states corresponded to an initial state with high empowerment. In the continuum, counting distinct states becomes meaningless. Therefore, here, we measure diversity of the final state by computing the variance of the flattened sample vectors of $S_f$, as summarized in Algorithm 1, and use this pragmatic approach directly in place of $\mathcal{E}$ in Eq. 1. While this proxy relies on an assumption of homogeneous noise and high SNR, it can be computed much more efficiently than empowerment, lending it to be directly applicable to human-in-the-loop training of assistive agents. Even as a coarse proxy for empowerment, we find that it is sufficient to significantly increase human success in our user study in Section 4.2. To empirically motivate our proxy, we compare the known empowerment landscape of a non-linear pendulum with our proxy result in Figure 2. A rigourous study of the properties of the proxy is deterred to the future work.

---

**Algorithm 1:** Empowerment-inspired Diversity Bonus

---

initialize environment at state $s_{init}$;
initialize empty list of final states $S_f$;
// rollout $N$ trajectories of horizon $T$
**for** $n = 1 \dots N$ **do**
    $s \leftarrow s_{init}$
    **for** $t = 1 \dots T$ **do**
        // randomly generate actions and update state
        $a$ = sample action;
        $s \leftarrow \mathcal{P}(s, a)$
    **end**
    $S_f \leftarrow S_f + [s]$
**end**
// To compute scalar reward bonus
**return** $\mathrm{Var}(\mathrm{flatten}(S_f))$

---

# 4  Experiments

In this section we evaluate our method in two distinct assistive tasks where a human attempts to achieve a goal: firstly, a shared workspace task where the human and assistant are independent agents, and secondly, a shared autonomy control task. The first task is used to directly compare the empowerment method in a tabular case against a goal inference baseline to motivate our method, and the second task is used to test both our proxy metric in Algorithm 1 and evaluate our overall method with real users in a challenging simulated teleoperation task. For accompanying code, see `https://github.com/yuqingd/ave`.

## 4.1  Shared Workspace – Gridworld

**Experiment Setup.** To motivate our method as an alternative when goal inference may fail, we constructed a gridworld simulation as follows: a simulated proxy human attempts to move greedily towards a goal space. The grid contains blocks that are too heavy for them to move, however, an agent can assist by moving these blocks to any adjacent open space. The agent observes the location of the blocks and the humans but not the location of the goal. As a metric for successful assistance, we measure the success rate of the human reaching the goal and the average number of steps taken.

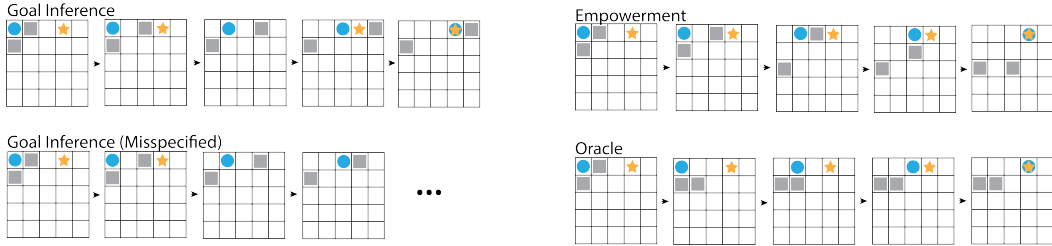

Figure 3: Sample rollouts with proxy human ●, immovable blocks ■, and goal at the star. Each frame consists of an action taken by the agent. Left column shows two cases of goal inference: ideal case (top) and failure mode with misspecified case (bottom) where the goal is blocked.

We compare against a goal inference method where the agent maintains a probability distribution over a candidate goal set and updates the likelihood of each goal based on the human's action at each step. We test variations of goal inference that highlight potential causes of failure: 1) when the goal set is too large, and 2) when the true goal is not in the goal set. Here, we define failure as the human's inability to achieve the goal before the experiment times out at 1000 steps. In this discrete scenario, we can compute empowerment directly by sampling 1000 trajectories and computing the logarithm of the number of distinct states the blocks and the human end up in. For comparison, we also provide results where we approximate empowerment with our proxy. The agent assumes that the human can move into any adjacent free space but does not know the human's policy or goal. We simulate a variety of initializations of the human's position, the number of blocks and their positions, and the goal position. Here we highlight the results of two main scenarios in Table 1: where the blocks trap the human in a corner, and where the blocks trap the human in the center of the grid. This is because the center of the grid has highest empowerment and the corner of the grid has lowest empowerment, and situations where the human is trapped are where assistance is most crucial. We ran 100 trials of randomized goal initializations and computed the number of steps the human takes to get to their goal (if successful) under each of the reward formulations.

**Analysis.** Our results suggest that in the case where we have a small and correctly specified goal set, goal inference is the best strategy to use as it consistently succeeds and takes fewer steps to reach the goal on average. However, this represents an ideal case. When we have a misspecified goal set and/or a larger goal set, the success rate drops significantly. The failure modes occur when the agent is mistaken or uncertain about the goal location, and chooses to move a block on top of the true goal. As the human can no longer access the goal, they wait in an adjacent block and cannot act further to inform the agent about the true goal location, leading to an infinite loop. Since our empowerment method does not maintain a goal set, it does not encounter this issue and is able to succeed consistently, albeit with a tradeoff in higher mean steps performance. Interestingly, we note that the failure cases that the goal inference method encounters could easily be resolved by switching to the empowerment method – that is, since the human is next to a block that is on top of the goal,

Table 1: Success rates and mean steps to goal (after removing failed trials) for human in corner (top) and human in center (bottom) scenarios. For human in corner, the human is randomly initialized in one of the four corners of the grid with two blocks trapping them. For human in center, the human is initialized at the center of the grid with four blocks surrounding them, one on each side. GI – goal inference with either the goal in the goal set (known) goal or the goal missing from the goal set (unknown). Large Goal (LG) Set considers every possible space as a goal, Small Goal (SG) Set only considered two possible goals, and No Goal (NG) is our method.

| Human in Corner \| Success %, (Mean Steps) | LG Set | SG Set | NG |
|---|---|---|---|
| GI (known) | 90%, (4.35) | **100%**, (4.35) | |
| GI (unknown) | 90%, (4.35) | 92%, (4.51) | |
| Empowerment | | | **100%**, (5.24) |
| Empowerment Proxy | | | **99%**, (8.79) |
| Oracle | | | 100%, (4.1) |
| Human in Center \| Success %, (Mean Steps) | LG Set | SG Set | NG |
| GI (known) | 50%, (2.27) | **100%**, (2.57) | |
| GI (unknown) | 50%, (2.27) | 55%, (3.3) | |
| Empowerment | | | **100%**, (4.8) |
| Empowerment Proxy | | | **100%**, (6.71) |
| Oracle | | | 100%, (1.91) |

increasing human empowerment would move the block away and allow the human to access it. This highlights a potential for future work in hybrid assistance methods. In assistance scenarios where accurate goal inference may be too challenging to complete or the risks of assisting with an incorrect goal are too high, our goal-agnostic method provides assistance while circumventing potential issues with explicitly inferring human goals or rewards. As motivation for our proxy, we note that the proxy increases the mean steps to goal and can lead to a 1% decrease in success rate. This is because the inaccurate measure of empowerment can occasionally lead to blocking the goal, as in the non-ideal goal inferences cases. However, the proxy success rate is still much higher than that of goal inference.

## 4.2 Shared Autonomy – Lunar Lander

Our previous experiments motivate the benefits of a goal-agnostic human assistance method using human empowerment. To evaluate our method with real users, we ran a user study in the shared autonomy domain. The purpose of this experiment is two-fold: we demonstrate that a simplified empowerment-inspired proxy metric is sufficient for assisting the human while avoiding the computational demands of computing empowerment, and we demonstrate the efficacy of our method by assisting real humans in a challenging goal-oriented task without inference. For clarity, in this section we use 'empowerment' to denote the proxy defined in Section 3.2.

We build on recent work in this area by [35] which proposed a human-in-the-loop model-free deep reinforcement learning (RL) method for tackling shared autonomy without assumptions of known dynamics, observation model, or set of user goals. Their method trains a policy (via Q-learning) that maps state and the user control input to an action. To maintain the user's ability to control the system, the policy only changes the user input when it falls below the optimal Q-value action by some margin $\alpha$. In this case, RL will implicitly attempt to infer the goal from the human input in order to perform better at the task, but this is incredibly challenging. With our method, empowerment is a way to explicitly encourage the shared autonomy system to be more controllable, allowing the user to more easily control the shared system. As a result, we hypothesize that optimizing for human empowerment will lead to a more useful assistant faster than vanilla RL.

**Experiment Setup.** The main simulation used is Lunar Lander from OpenAI Gym [8], as it emulates challenges of teleoperating a control system. Generally human players find the game incredibly difficult due to the dynamics, fast reaction time required, and limited control interface. The goal of the game is to control a lander using a main thruster and two lateral thrusters in order to land at a randomly generated site indicated by two flags, giving the player a positive reward. The game ends when the lander crashes on the surface, exceeds the time limit, or flies off the edges of the screen, giving the player a negative reward. The action space consists of six discrete actions that are combinations of the main thruster settings (on, off) and the lateral thruster settings (left, right, off). To assist players, we use DQN to train a copilot agent whose reward depends on landing successfully

at the goal, but cannot observe where that goal is and instead keeps a memory of the past 20 actions input by the user. The user input and copilot action space are identical and the state-space contains the lander's position, velocity, tilt, angular velocity, and ground contact sensors. We compute the empowerment-based reward bonus by randomly rolling out 10 trajectories at each state and computing the variance in the final positions of the lander, as described in Section 3.2. Since both the human and agent control the same lander, the empowerment of the shared autonomy system is coupled. We hypothesize that including the empowerment term would be particularly useful for a challenging control task such as Lunar Lander, since states where the lander is more stable are easier for the player to control.

**Simulation Experiments.** We first test our method using simulated 'human' pilots as designed in the original study [35]. First, an optimal pilot is trained with the goal as a part of its state space using DQN. Each of the 'human' pilots are imperfect augmentations of this optimal pilot as follows: the Noop pilot takes no actions, the Laggy pilot repeats its previously taken action with probability $p = 0.85$, the Noisy pilot takes a wrong action with probability $p = 0.3$ (e.g. go down instead of up), and the Sensor pilot only moves left or right depending on its position relative to the goal, but does not use the main thruster. We use these pilots as simulated proxies for humans and train two copilots with each of them: one with empowerment added to the reward function and one without. For the empowered copilots, we do a hyperparameter sweep over $c_{emp}$ from 0.00001 to 10.0 and found the best performance with $c_{emp} = 0.001$. The copilots were trained on 500 episodes of max length 1000 steps on AWS EC2. We then conduct a cross evaluation for each copilot and pilot pair by counting successful landings across 100 evaluation episodes, averaged across 10 seeds.

Table 2: Best successful landing percentages for each simulated pilot in 100 episodes, averaged across 10 seeds. Our method improves upon the baseline for all pilots except Laggy, where the performance is on par with the baseline.

|  | Full | Noop | Laggy | Noisy | Sensor |
|---|---|---|---|---|---|
| No Copilot | **58.2** | 0 | 11.9 | 11.6 | 0 |
| No Empowerment (Baseline) |  | 5.9 | **38.4** | 8.5 | 2.8 |
| Empowerment |  | **9.7** | 37.3 | **30.7** | **10.8** |

We find that for almost all simulated pilots, our method leads to increased successful landing rate (shown in Table 2) when paired with the best copilot. In the only case where we do not outperform the baseline, with the Laggy pilot, the success rates between the baseline and our method only differ by 1.1%. In all cases, empowerment perform better than the simulated single pilot with no copilot. From our simulation results we see that our method can increase controllability for these simulated pilots, leading to higher successful landing rates in Lunar Lander.

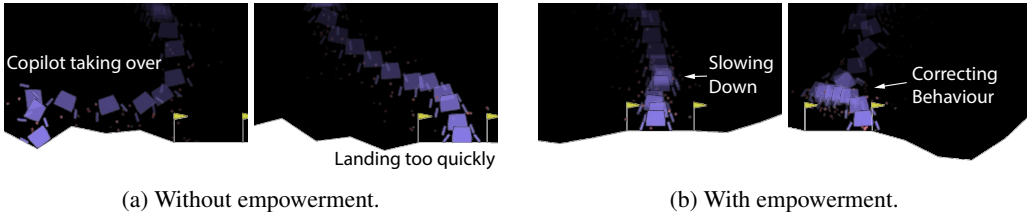

(a) Without empowerment.　　　　　　　　　　(b) With empowerment.

Figure 4: Sample trajectories from the user study. The leftmost image shows the copilot take over and swing the lander to the left even though the user is trying to move right. On the second image from the left, the copilot does not slow the motion down sufficiently so the lander has too much momentum and fails to land on the legs, leading to a crash. In the two right images, we see the frames overlap more than in the no empowerment case due to the slower, more stable motion. The rightmost image shows corrective stabilization behaviour when approaching the goal from an inconvenient angle. See `https://youtu.be/oQ1TvWG-Jns` for a video summary of user study results.

**User Study.** To test our method with real pilots, we conducted an IRB-approved user study to compare human player performance on Lunar Lander with human-in-the-loop training. As with the simulated experiments, we manipulate the objective the copilot is trained with: our method with the empowerment bonus in the reward function ($c_{emp} = 0.001$) and the baseline with no empowerment. We found that the quality of assistance depends on this coefficient as follows: increasing $c_{emp}$

generally makes the copilot more inclined to focus on stabilization, but if $c_{emp}$ is too high, the copilot tends to override the pilot and focus only on hovering in the air. The objective measure of this study is the successful landing rate of the human-copilot team, and the subjective measure is based on a 7-point Likert scale survey about each participant's experience with either copilot. We designed the survey to capture whether the copilots improved task performance, increased/decreased the user's autonomy and control, and directly compare the user's personal preference between the two copilots (refer to Table 4). We recruited 20 (11 male, 9 female) participants aged 21-49 (mean 25). Each participant was given the rules of the game and an initial practice period of 25 episodes to familiarize themselves with the controls and scoring, practice the game, and alleviate future learning effects.

To speed up copilot learning, each copilot was first pretrained for 500 episodes with the simulated Laggy pilot, then fine-tuned as each human participant played for 50 episodes with each of the two copilot conditions without being informed how the copilots differ. To alleviate the confounding factor of improving at the game over time, we counterbalanced the order of the two conditions.

The objective success rates from the user study are summarized in Table 3. We ran a paired two-sample t-Test on the success rate when the copilot was trained with and without empowerment and found that the copilot with empowerment helped the human succeed at a significantly higher rate ($p < 0.0001$), supporting our hypothesis.

Table 3: Objective User Study Results: Average success percentages with standard deviation.

|  | No Empowerment | Empowerment |
|---|---|---|
| Success Rate | $17 \pm 9$ | $\mathbf{33 \pm 7}$ |

The results of our survey are summarized in Table 4 where we report the mean response to each question for each copilot and the p-value of a paired two-sample t-Test. To account for multiple comparisons, we also report the Bonferroni-corrected p-values. We find that the participants perceived the empowerment copilot's actions made the game significantly easier ($p = 0.007$) as compared to the no-empowerment copilot. Furthermore, the two comparison questions have a significant level of internal consistency, with Chronbach's $\alpha = 0.96$, and the most frequent response is a strong preference (7) for the empowerment copilot. Although on average the perception of control and assistance is higher with the empowerment copilot, this difference was not found to be significant. Comments from the participants suggest that the empowerment copilot generally provided more stabilization at the expense of decreased human left/right thruster control, which the most users were able to leverage and collaborate with – the copilot increased stability and reduced the lander speed, allowing the user to better navigate to the goal (see Figure 4b). On the other hand, the no-empowerment copilot did not consistently provide stability or would take over user control, moving away from the goal (see Figure 4a). These results are exciting as prior work in shared autonomy has proposed heuristics for assistance, such as virtual fixtures [28, 1] or potential fields [3, 9], but we find that one benefit of empowerment is the natural emergence of stabilization without relying on heuristics. We also note that our empowerment-inspired reward bonus allowed us to leverage the intuitive benefits of controllability without the large computational costs of the empowerment method, allowing the empowerment-based copilot to learn alongside each human player in real time. While our method empirically demonstrated the merits of empowerment-based assistance, the user study also highlighted limitations of assistance through pure empowerment. Predominantly optimizing for empowerment can lead to failure modes where the assistant prevents landing altogether due to prioritizing stability excessively.

## 5 Discussion and Conclusion

**Summary.** In this work, we introduce a novel formalization of assistance: rather than attempting to infer a human's goal(s) and explicitly helping them to achieve what we think they want, we instead propose empowering the human to increase their controllability over the environment. This serves as proof-of-concept for a new direction in human-agent collaboration. In particular, our work circumvents typical challenges in goal inference and shows the advantage of this method in different simulations where the agents are generally able to assist the human without assumptions about the human's intentions. We also propose an efficient algorithm inspired by empowerment for real time

Table 4: Subjective User Study Survey Results. 1 = strongly disagree, 7 = strongly agree. **p-value**$^*$ are Bonferonni-corrected p-values.

| | Question | Emp | No-Emp | p-value | p-value$^*$ |
|---|---|---|---|---|---|
| Autonomy | I had sufficient control over the lander's movement. | 4.7 | 3.6 | **0.025** | 0.175 |
| | The copilot often overrode useful controls from me. | 4.5 | 5.35 | 0.053 | 0.371 |
| | The copilot did not provide enough assistance. | 3 | 3.45 | 0.14 | 0.98 |
| | The copilot provided too much assistance. | 3.15 | 3.95 | 0.12 | 0.84 |
| | The quality of the copilot improved over time. | 6.25 | 5.75 | 0.096 | 0.672 |
| Perform | The copilot's actions made parts of the task easier. | 6.4 | 5.55 | **0.001** | **0.007** |
| | The copilot's assistance increased my performance at the task. | 6.4 | 5.6 | **0.032** | 0.21 |

| Comparison Questions | Chronbach's $\alpha$ | Mean | Mode |
|---|---|---|---|
| I preferred the assistance from the empowerment copilot to the no empowerment copilot. <br> I was more successful at completing the task with the empowerment copilot than the no empowerment copilot. | 0.96 | $4.9 \pm 2$ | 7 |

assistance-based use cases, allowing us to conduct human-in-the-loop training with our method and demonstrate success in assisting humans in a user study.

**Limitations and Future Work.** Our experiments find that in cases where the human's goals can be inferred accurately, general empowerment is not the best method to use. As there exist situations where optimizing for human empowerment will not be the best way to provide assistance, in future work we seek to formalize how goal-agnostic and goal-oriented assistance can be combined. For example, a natural continuation of this work we will explore a hybrid approach, combining local and global planning. Another area of future work is to explore the use of human empowerment for assisting humans with general reward functions. In this paper, we primarily focus on assisting with goal states as a way to compare with existing goal inference assistance methods, but the human empowerment objective can potentially apply to more general reward formulations. Furthermore, we proposed a proxy to empowerment in order to make a user study feasible, which came at a cost of not being an accurate measure of true empowerment. Although we found that our simplified method led to significant improvement in our user study, the current proxy assumes homogeneous noise and is sensitive to scenarios with noise varying between different states, and the sample-based method naturally requires more computational power as action space grows – that being said, there are many meaningful assistance applications with small action spaces (e.g. navigation with mobility devices, utensil stabilization). Future work can analyze the tradeoffs between computational tractability and numerical empowerment accuracy in the human assistance domain.

## Broader Impact

As our work is focused on enabling artificial agents to learn to be more useful assistants, we believe it has the potential for significant broader impact in both the research community and for the future usefulness of assistive agents in a variety of real world applications (e.g. assistive robotics in elder care, prosthetics). The most immediate impact of our work lies in the direct application of empowerment for assisting humans. In the research community, the novel use of empowerment for human assistance can motivate further work in goal-agnostic human-agent collaboration and assistance. As emphasized in our work, in cases where goal inference is challenging or when aiding under the assumption of an incorrectly inferred goal may have risks, our method acts as an alternative to potential pitfalls of goal inference. This, we believe, can be crucial when applied to assisting people in the real world. In particular, our successful results with the Lunar Lander user study suggests that our method can assist humans engaging in a challenging shared autonomy control task. Our method invites extensions to real-world shared autonomy tasks (e.g. flying a quadrotor drone, teleoperation of a high DOF robotic arm with a joystick controller, etc.). Outside of research labs, the broader impact of a goal-agnostic human assistive method lies in the potential of applying this general method to a wide variety of assistive tasks – ranging from software assistants to assitive robotics. We also emphasize that while assistive technologies are developed to aid people, safety procedures and strict certification are necessary before a real-world application of our method. With direct human interaction, failures of the system can critically impact people (whether that be through physical robotic failures, or privacy concerns with software assistants).

At a societal scale, we also hope that proposing a method that optimizes for *human controllability* will encourage future ethical discussions about how to realize learning assistive agents that balance providing effective help while also ultimately guaranteeing as much human autonomy and control as possible. Importantly, as empowerment aims to enhance the element of *autonomy* in the human, this offers a systematic route to avoid the possible drawback of an overly helpful, but constricting artificial assistant. Depending on the situation in which AI agents are employed, there can be uncertainty around the extent to which people (at a personal level, cultural level, etc.) require a balance between autonomy and assistance. The importance of different types of autonomy (e.g. personal, moral) for different groups of individuals (e.g. age groups, cultural groups) and how they can be positively or negatively affected by applications of human empowerment can be examined in other areas of research (e.g. sociology, philosophy, psychology).

## Acknowledgements

This research is supported by: an Azure sponsorship from Microsoft, a Berkeley AI Lab (BAIR) Fellowship, ONR through PECASE N000141612723, and NSF under grant NRI-#1734633.

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
