[Supplementary Material · AvE_Supplementary_Material.pdf]

# AvE Supplementary Material

## 1    Additional Gridworld Experiments

In the main paper we highlight the most interesting scenarios where assistance is necessary: where the human is trapped by immovable boxes in either the corner or center of the grid. We also tested other initializations as follows:

| Human in Corner — Success %, (Mean Steps) | LG Set | SG Set | NG |
|---|---|---|---|
| GI (known) | 87%, (1.79) | **100%** , (1.79) | |
| GI (unknown) | 86%, (1.79) | 88%, (1.82) | |
| Empowerment | | | **100%**, (2.13) |

Table 1: Random initialization of the human's position, the goal's position, and the position of two blocks. Results averaged across 100 trials. GI – goal inference with either a (known) goal in the goal set or an (unknown) goal missing from the goal set. Large Goal (LG) Set considered every possible space as a goal, Small Goal (SG) Set only considered two possible goals, and No Goal (NG) is our method.

| Human in Corner — Success %, (Mean Steps) | LG Set | SG Set | NG |
|---|---|---|---|
| GI (known) | 89%, (3.25) | **100%** , (3.25) | |
| GI (unknown) | 89%, (3.25) | 96%, (3.25) | |
| Empowerment | | | **100%**, (3.73) |

Table 2: Human randomly initialized in any corner and goal randomly initialized. Blocks initialized to be one space away from trapping the human. Results averaged across 100 trials. GI – goal inference with either a (known) goal in the goal set or an (unknown) goal missing from the goal set. Large Goal (LG) Set considered every possible space as a goal, Small Goal (SG) Set only considered two possible goals, and No Goal (NG) is our method.

## 2    User Study Results

Please see the attached `AvE User Study Results Video.mp4` for recorded clips from the user study.

I was more successful at completing the task with empowerment copilot than no empowerment.

I preferred the assistance from empowerment copilot to no empowerment.

No empowerment copilot's assistance increased my performance at the task.

Empowerment copilot's assistance increased my performance at the task.

No empowerment copilot's actions made parts of the task easier.

Empowerment copilot's actions made parts of the task easier.

The quality of no empowerment copilot improved over time.

The quality of empowerment copilot improved over time.

No empowerment copilot provided too much assistance.

Empowerment copilot provided too much assistance.

No empowerment copilot did not provide enough assistance.

Empowerment copilot did not provide enough assistance.

No empowerment copilot often overrode useful controls from me.

Empowerment copilot often overrode useful controls from me.

With no empowerment, I had sufficient control over the lander's movement.

With empowerment, I had sufficient control over the lander's movement.

4 = Neutral    3    2    1 = Strongly Disagree    5    6    7 = Strongly Agree

Figure 1: Survey responses from Lunar Lander User Study