[Reviews · NeurIPS 2020]

Review 1

Summary and Contributions: The authors contribute a goal-agnostic method through which an assistive agent can improve joint task performance. The insight provided in this work is the inclusion of the variance of potential outcomes (given a horizon) in the assistive agent's reward function. The work is evaluated in an illustrative gridworld domain and the lunar lander domain.

Strengths: The primary strength of this work is that their method appears to show promise on the domains they've applied it to, leading to improved performance as compared to baseline methods that utilize goal inference (or to methods from closely related work in shared autonomy [35]). The authors conducted a user study that showed users perceived their method's utility: participants found the assistive agent's actions helpful when using the presented proxy for empowerment.

Weaknesses: The primary weakness of the work is that the effects of the new reward function term are not rigorously characterized, but instead empirically demonstrated on two domains with very small action spaces. As the authors elaborate, computing the actual empowerment value of a state is not feasible, and so instead they perform random limited-horizon rollouts and compute variance across the final state vectors from each. It would strengthen the paper to include details about how the state space formulation impacts this proxy, how much this proxy sacrifices versus the actual computation of empowerment (using a non-realtime scenario with a trained agent), how action space size impacts the applicability of the proposed method, and a more thorough treatment of how c_{emp} impacts behavior, particularly since this hyperparameter appears to be critical for properly incorporating this new reward signal and not harming human/lead agent performance.

Correctness: The authors' empirical methodology is sound, as is their analysis. It seems strong to call their contribution an application of empowerment given the difference between their proxy and the initial definition of empowerment.

Clarity: The paper is very well written and easy to follow, a great example of a good idea executed and presented cleanly.

Relation to Prior Work: The contribution, while using a nearly identical evaluation methodology to recent prior work, is sufficiently differentiated by its underlying method.

Reproducibility: Yes

Additional Feedback: The authors' rebuttal was well crafted and addressed a concern regarding the fidelity of the empowerment proxy, focusing on a 2D navigation setting (and gridworld-analogous environments). It remains unclear how this approach will scale to more complex state spaces, but I believe that if the contributions are appropriately scoped in description that this paper is above the acceptance threshold.


Review 2

Summary and Contributions: The paper describes a novel method for intrinsic reinforcement learning in which a measure of "empowerment" of a human is added to a reward function with a scaling factor. A variational approximation is proposed, as well as a second sampling-based approximation. An agent acting on the reward function so defined will generate policies that favor helping a human in some way (increasing the human's empowerment, an existing concept) over policies that achieve a goal. Experiments are comprehensive on both simulation toy and real domains and on human users in a real domain.

Strengths: - a well defined construct of empowerment - Comprehensive evaluation of the construct in both simulations of toy and real world domains with humans. - The evalutions are really quite impressive.

Weaknesses: - The relationships to the literature I describe below. - the overall novelty of the proposed algorithm

Correctness: seems correct

Clarity: very well written modulo figure 1. I don't see how Figure 1 shows what is stated at the end of the first paragraph. This could use some clarification

Relation to Prior Work: Altruism rewards such as this have been investigated before, however, so I don't see what this paper really brings to the table other than the connection of empowerment to the channel capacity. It seems this connection has already been made (e.g. ref [23]) so the novelty of what is left is somewhat questionable. Recent work by Jaques et al use a measure of "influence" and agent has on other agents, which seems related to the empowerment measure. See also Ray et al 2008 and Moutoussis et al 2014 who used a measure of power and esteem to modulate a reward function. I'm also somewhat surprised to not see a reference to the free energy of a system, and the ensuing active inference model which seems to have many similarities to the measure of empowerment proposed. The basic idea of a Bayesian reinforcement learning algorithm, for example, is exactly to optimize an action sequence over a set of preferred future states by modulating a transition, observation and reward function. I feel a connection to this literature is lacking in the current paper. This recent preprint describes empowerment in a way that is very similar to the one proposed by this submission, but I would not expect the authors to have seen as it was published online after the NeurIPS deadline. https://arxiv.org/abs/2006.04120

Reproducibility: Yes

Additional Feedback: I'm very supportive of this work, as I think it is taking AI in a very promising direction. My comments about related work are mostly there (I hope) to strengthen the work. Nevertheless, the overall novelty of this particular approach is not as clear as it could be. Also, the video is hilarious but it would be even better if the freeze-frame titles describing what method is currently being used are kept as titles somewhere during the subsequent video clips so people like me can remember what type we are watching more easily. Also the video could be shortened by removing some of the redundant crash-landings (which are hilarious for some reason, I think because I'm picturing the poor human as the agent takes over and crashes the lander. Its interesting to think how this may have affected future play). One thing that is not covered in the "broader impact" statement is the notion that the assistive agent actions can change the human behaviour. An overly agressive agent (the parameter c_emp again I believe) may lead to a change in human behaviour. Manipulation of this level of aggressiveness could lead to human manipulation. Something to consider in any case. Here are some additional, more detailed comments The distinction between actual empowerment method and the empowerment proxy is somewhat confusing. The paper seems to apply claims across both methods despite the two methods being somewhat distinct. For instance, they use the results from the second experiment as if they apply to the overall empowerment method, despite it testing the proxy which makes me somewhat dubious. From the second experiment, the paper makes the claim that co-pilots learn alongside human players in real-time, but I don’t think this claim is necessarily supported. The paper claims that each pilot was pretrained with the simulated Laggy pilot (line 262), but I couldn’t see any information about this pretraining. Additionally, the paper then claims that each co-pilot learns from the user during their 50 episodes of the experiment. My thinking here is that if the co-pilots are pre-trained 500 episodes with the simulated human (as in the simulation experiments), do the co-pilots actually learn at all during their only 50 episodes with actual humans? (Though I suppose one could try and claim that training for 50 episodes, whether the agent’s policy is updated or not, counts as learning.) Some comparison between the “pretrained co-pilots” and the “fine tuned on human pilot data co-pilots” could be interesting, but not necessarily relevant to this paper. Details about the pretraining should be added. A paragraph in Section 2 Empowerment Preliminary (lines 85 to 96) is written strangely. It talks about reference [40], switches to talking about reference [17], but then swings back to talking about reference [40] again as if it wasn’t already mentioned. I found it to be confusing and it could easily be rearranged to make it clearer. There is some kind of mistake in the sentence starting on line 335 and ending on line 337 that reads “Depending on the situation in which AI agents are employed, there can be uncertainty around the extent to which people (at a personal level, cultural level, etc.) are como a balance between autonomy and assistance.” I’m not sure what it’s trying to say. The authors did not address my concerns or questions in the rebuttal.


Review 3

Summary and Contributions: This paper presents the notion of embedding empowerment, i.e. increasing the number of future reachable states, into an RL framework for an agent to provide assistance to a human in a task. The contributions are: a straightforward (in a positive sense) RL objective which includes a weighted mixture of a task reward and an empowerment reward; a proxy metric for empowerment in for continuous domains; and an interesting user study where the new method is trialled on Lunar Lander with real human players.

Strengths: 1) The clarity of vision of the proposed work and the simplicity of the mechanism design. This makes the paper very easy to read and understand, and the results easy to interpret. 2) The paper is very well written and a pleasure to read. 3) The user study is a great inclusion for this paper, and not something I've seen in many other papers. 4) Whilst neither RL for human assistance nor empowerment are new concepts, the combination here is a good contribution that has the potential to spawn many follow-ups. So although it may not be highly novel, I think it has the potential to be quite significant.

Weaknesses: 1) The proxy for empowerment feels like an engineering solution. This is fine, but it's not explored in any real detail, and we never see a comparison between the proxy and the true empowerment value. 2) The evaluation is over two domains, one of which is relatively simple in terms of the human objective, and in both cases operate in fairly straightforward physical environments. The control in the Lunar Lander is definitely not simple, but in both environments increasing the number of reachable states directly translates to goal achievement probability. Are there other environments with hazards or obstacles that could present a tougher challenge? 3) How RL features in the gridworld experiment is not clear in the paper.

Correctness: Yes, all looks correct to me.

Clarity: The paper is extremely well written.

Relation to Prior Work: The prior work is well presented and the differences are clear. The most relevant work in terms of human assistance is used as a baseline.

Reproducibility: Yes

Additional Feedback: Figure 1 is a very odd example it made very little sense to me, and was harder to grasp than both of your empirical domains. p 3 / line 80 doesn't parse well around S_T. line 136 states the same thing twice and is "quite" informal. Is there some complexity analysis to call on here? Figure 2 - the blue circles are different colours in the figure and caption As a minor aside, in https://arxiv.org/abs/1911.04848 there is an interesting example of the problem of goal inference in shared autonomy. The assitive controller in that paper causes problems (I think they call it conflict of control) when the human's teleoperation goal and the robot's model of their goal is different. After discussion with other reviewers, I am downgrading my score slightly. This is still a paper I would argue to accept, but I find that the lack of a more rigorous exploration of the new approach to be more of a problem with hindsight and some additional context. The promised changes in the rebuttal (e.g. evaluation of proxy on the gridworld) add further support for acceptance, and more formal work in this direction would also help.


Review 4

Summary and Contributions: The paper addresses the problem of constructing agents that act as assistants for human users. The authors circumvent the challenge of inferring the user's goals by instead training the agent to increase the amount of controllability of the user over their environment. For this, the agent's reward is augmented with a term called empowerment, which is a measure for the number of future states the human can reach from its current state. The paper further proposes a computationally simpler version of this objective and presents experiments, including a user study.

Strengths: The problem addressed is certainly relevant to the NeurIPS community. The empowerment approach appears to be quite general and widely applicable with few modifications. The user study appears to be sound and provide significant results.

Weaknesses: The properties and limitations of the proposed empowerment proxy should be discussed in more detail (see "additional feedback"), if possible formally. Reproducibility of the experiments could be improved: in section 4.1, the authors "compare against a goal inference method where the agent maintains a probability distribution over a candidate goal set and updates the likelihood of each goal based on the human’s action at each step." How exactly does the baseline method do this?

Correctness: The empirical methodology of the user study and the experiments with simulated users seem to be correct. I have some questions regarding the conclusions drawn from the experiements (see "additional feedback").

Clarity: The paper is reads easily.

Relation to Prior Work: I am unfortunately not very familiar with related work in this area.

Reproducibility: Yes

Additional Feedback: I would appreciate a discussion of the potential limitations of the presented empowerment proxy (Alg. 1): What about environments where different states reachable from the current state have different amounts of environmental noise? Consider for example crossing a street on a pedestrian crossing vs. jaywalking. The latter has many different possible outcomes while the former almost certainly just results in reaching the other side. If a wheel-chair agent tries to "empower" its human occupant with the presented proxy, what would happen? If I read [38] correctly, it seems that the sparse sampling approximation the authors take inspiration from is only applicable for deterministic environments, yet the authors make no such statement regarding their definition. It would be good to clarify any such limitations. Similarly, in the lunar lander environment: Doesn't the proposed empowerment proxy incentivize the agent to *prevent* the landing? The closer to the goal you are, the more often a trajectory will end at the actual goal, i.e., lead to a small variance in the set of final states. Elaborating a little further here, the agent reward in the lunar lander experiment seems to be the true reward (landing at the goal) plus the empowerment bonus, with the added complication that the agent cannot observe where the goal is. It does however see the last 20 actions of the human. This enables the agent to try to infer the goal location from the user's actions. To me, this seems to be a combination of goal-agnostic and goal-oriented assistance. Yet the authors state that such a combination is future work. If it isn't such a combination, why is landing included in the reward at all? Can the authors comment on this? It seems paper [40] is very close and essentially proposes the same idea. In that context, I don't understand the authors' statement "These approaches have shown the applicability of empowerment to human-agent collaborative settings, however, these empowerment-only goal agnostic approaches are not applicable directly to assistance scenarios where one player, the human, has a certain goal." Indeed, the cited paper states in its discussion section: "Empowerment endows a state space cum transition dynamic with a generic pseudo-utility function that serves as a rich preference landscape without requiring an explicit, externally defined reward structure; on the other hand, where desired, it can be combined with explicit task-dependent rewards." Can the authors elaborate here? In the Shared Workspace Gridworld experiment, it would be interesting to compare both the empowerment definition in equation (1) or the proxy from algorithm (1), in order to understand the magnitude of approximation introduced by moving away from equation (1). Update after feedback phase: my questions were mostly answered, and the authors also had convincing answers to my concerns (e.g., in the case of the additional experiment with the proposed proxy on the gridworld benchmark and its convincing result). As they promise to address these points in the final version, I see no obstacle to raising my overall score above the acceptance threshold.

[Author Response · NeurIPS 2020]

We thank the reviewers for their detailed and insightful feedback! We are happy that the reviewers find our problem relevant and our evaluation and user study sound and significant. The reviewers recognize that we introduce a novel formulation of the assistance problem—tested for the first time with humans—with significant potential to spawn further future work in this direction. Due to space constraints, we focus our author response on addressing R1 and R4's core concern, comparing our proxy against true empowerment, and also provide some clarifications.

**Empirical comparisons of the proxy with empowerment (R1, R3, R4):** As it is not computationally feasible to directly compute empowerment in most domains, especially in real-time with humans in the loop, we motivate our proxy with experiments in domains where computing empowerment is feasible: 1) the known empowerment landscape of the non-linear pendulum 2) as suggested by R4, we evaluate the proxy method in the gridworld.

Following the intuition of the empowerment as quantifying the diversity of future states, our proxy estimates diversity by variance of final states. We find empirically in the well-known testbed of a non-linear pendulum that our proxy captures the essential properties of the empowerment landscape: the maximum is at the upright position of the pendulum (center of plot) and empowerment values are comparatively low for the states with energy below the separatrix energy. The

left plot is the landscape of the proxy and the right plot is the corresponding empowerment landscape. Importantly, we do not aim to reproduce exact empowerment values, but rather to generally capture the analogous relative changes. The proxy indeed captures the critical qualitative features.

For the gridworld, replicating the experiments using the proxy instead of empowerment leads to a 1% decrease in success rate. Examining the failed cases shows that the proxy's inaccurate measure of empowerment can occasionally lead to blocking the goal, as in the non-ideal goal inference cases, however, the proxy failure rate is still much lower than that of goal inference (1% vs. 18%). These results suggest that the proxy appears to be a practical replacement for empowerment for the tested scenarios. Emphatically, the underlying intuition for our approach is to offer humans increasing controllability online; thus, computational efficiency is more critical than highly accurate empowerment.

**Limitations of the proxy and of empowerment (R1, R4):** We agree with the reviewers' suggestions that the paper would be strengthened with a more detailed treatment of the limitations of the proxy when used for assistance. We note that using the variance as a proxy for empowerment relies on an the assumption of homogeneous noise and high SNR, and would like to acknowledge R4's important insight that scenarios with noise varying between states would mislead our proxy, and will add these limitations to our discussion. To address R1's scalability comments, the proxy's sample-based method naturally requires exponentially more computational power as the action space grows. Nevertheless, there are many meaningful assistance applications with small action spaces (e.g. navigation with mobility devices, utensil stabilization). For large action spaces, we envisage that a natural way to achieve computational efficiency is by focusing on a subset of the action space most relevant for the range of tasks, which is an exciting area for future work. As for limitations with empowerment, as R4 suggests, predominantly optimizing for empowerment can lead to failure modes where the assistant prevents landing altogether (shown in the supplementary video); we will emphasize this point. R3 asks if there are environments that challenge/mislead empowerment. As our discussion acknowledges, we do not claim that empowerment alone is a general solution to all assistive tasks. An example of this is when the goal of the human requires them to lose empowerment to reach it, e.g. when the goal is inside a tunnel.

**Novelty of empowerment-based assistance (R4, R2):** We clarify: while prior work has used empowerment in agent rewards, we are proposing a formalism for assistance that is new relative to existing assistance formulations. As R3 mentions, although the components of our method are not novel, using them in this way is. R4 asks how our approach differs from [40], and we clarify that while [40] conceptually discusses the idea of an agent optimizing for human empowerment as it relates to Asimov's 3 laws of robotics, our paper concretely proposes that assistance in common shared workspaces and shared autonomy tasks can be cast as an empowerment problem, and evaluates that idea with real users. We certainly took inspiration from [40] in our formulation, and will amend the statement accordingly.

**Combination of goal-agnostic and goal-oriented assistance (R4):** To clarify, R4 is correct that allowing the agent to observe the recent human actions will lead the agent to implicitly try to infer the goal location. However, this does not rely on an explicit candidate set of goals, and the agent might only implicitly run inference over some lower-dimensional latent space (whatever increases success). The landing reward is necessary in the shared autonomy case as we discuss in the above limitations: solely focusing on empowerment can lead to failed assistance. The future work we discuss hopes to combine the strengths of goal-inference assistive methods with goal-agnostic assistance, as there are cases where the strengths of one can overcome the weaknesses of the other and vice versa.

**Effect of $c_{emp}$ (R1):** In the Lander studies, we find that increasing $c_{emp}$ generally makes the copilot more inclined to focus on stabilization, but too high $c_{emp}$ causes the copilot to override the user and focus on hovering in the air as it prioritizes the high empowerment state over the reward of actually reaching the goal. We will add this to the discussion.

[Meta-Review · NeurIPS 2020]

Following the authors' response and the discussion all reviewers agree that the paper should be accepted at NeurIPS. The paper proposes a nice formulation for human-computer collaboration. The proposed technique is clearly explained and shows good performance on both a simulated toy domain and a human user study. While there are some clear limitations (experiments are performed on simplistic domains, assumes homogeneous noise and high SNR, etc.), the paper is definitely a good step forward and is likely to inspire followup work.